# Chromosome-Level Comprehensive Genome of Mangrove Sediment-Derived Fungus *Penicillium variabile* HXQ-H-1

**DOI:** 10.3390/jof6010007

**Published:** 2019-12-23

**Authors:** Ling Peng, Liangwei Li, Xiaochuan Liu, Jianwei Chen, Chengcheng Shi, Wenjie Guo, Qiwu Xu, Guangyi Fan, Xin Liu, Dehai Li

**Affiliations:** 1BGI-Qingdao, BGI-Shenzhen, Qingdao 266555, China; pengling@genomics.cn (L.P.); liliangwei@genomics.cn (L.L.); Liuxiaochuan@genomics.cn (X.L.); chenjianwei@genomics.cn (J.C.); shichengcheng@genomics.cn (C.S.); guowenjie@genomics.cn (W.G.); xuqiwu@genomics.cn (Q.X.); fanguangyi@genomics.cn (G.F.); 2China National GeneBank, BGI-Shenzhen, Shenzhen 518120, China; 3Key Laboratory of Marine Drugs, School of Medicine and Pharmacy, Ocean University of China, Qingdao 266100, China

**Keywords:** *Penicillium variabile*, chromosome-level genome, mangrove, secondary metabolites, phylogeny

## Abstract

*Penicillium* is an ascomycetous genus widely distributed in the natural environment and is one of the dominant fungi involved in the decomposition of mangroves, which can produce a variety of antitumor compounds and bioactive substances. However, in mangrove ecosystems there is no complete genome in this genus. In this study, we isolated a fungus strain named *Penicillium variabile* HXQ-H-1 from coast mangrove (Fujian Province, China). We generated a chromosome-level genome with total size of 33.32 Mb, scaffold N50 of 5.23 Mb and contig N50 of 96.74 kb. Additionally, we anchored about 95.91% assembly sequences into the longest seven scaffolds, and predicted 10,622 protein-coding genes, in which 99.66% could be annotated by eight protein databases. The secondary metabolites analysis reveals the strain has various gene clusters involving polyketide synthase (PKS), non-ribosomal peptide synthetase (NRPS) and terpene synthase that may have a largely capacity of biotechnological potential. Comparison genome analysis between *Penicillium variabile* and *Talaromyces islandicus* reveals a small difference in the total number of genes, whereas HXQ-H-1 has a higher gene number with COG functional annotation. Evolutionary relationship of *Penicillum* based on genome-wide data was carried out for the first time, showing the strain HXQ-H-1 is closely related to *Talaromyces islandicus*. This genomic resource may provide a new resource for development of novel bioactive antibiotics, drug candidates and precursors in *Penicillium variabile*.

## 1. Introduction

*Penicillium* is widely distributed in worldwide. It is known for its asexual forms and is found to inhabit soil, water, marine, mangrove swamps, decaying vegetation, air. Further *Penicillium* is a common contaminant of various substances. It plays an important role in human life for the function of decomposing various organic substances and produces antibiotics, toxic compounds (mycotoxins), extracellular enzyme, and antitumor active secondary metabolites [1,2,3]. Many species of *Penicillium* have been widely used in biotechnology, food, pharmaceutical, biological degradation, and many other applicable fields.

Marine-derived *Penicillium* has provided enormous excellent pharmaceutical leads over the past decades [3]. The manglicolous fungi represent the second largest marine fungi ecological group [4], and play an important role in mangrove ecosystems as the main decomposer and participant in the energy flow of mangroves, having significant effects on the survival, growth, and fitness of mangroves. In addition, secondary bioactive metabolites are derived from mangrove fungi, specially from mangrove endophytic fungi, which are used in pharmaceutical and nutraceutical industries to produce antimicrobial, anticancer, antioxidant, antidiabetic, and other therapeutic agents [5]. Here, we isolated a *Penicillium variabile* strain named HXQ-H-1 derived from mangrove. Multiple novel bioactive molecules were isolated from HXQ-H-1 by co-cultivation of *P. variabile* with *Talaromyces aculeatus* and DNA methyltransferase inhibitor respectively, of which nafuredin B and varitatin A were detected has the activity of anti-cancer cells [6,7]. The *P. variabile* strains derived from other environments have also been verified to have different metabolites. It’s reported that *P. variabile* Sopp 1912 derived from permafrost deposits can synthesize clavine alkaloids, rugulovasines A and B which was toxic to animals and plants [8,9,10], *P. variabile* JN-A525 with urethanase activity which could directly catalyze the decomposition of urethane in wine [11]. *P. variabile* P16 produce glucose oxidase which was used for catalyzing the oxidation of glucose in industry [12]. Closely related species of *P. variabile* have been investigated at the genomic level, such as *Talaromyces islandicus* (“*Penicillium islandicum*”) assembly genome with scaffold N50 of 2.35 Mb [13] and *Penicillium digitatum* assembly genome with scaffold N50 of 1.53 Mb [14]. *P. digitatum* has demonstrated an inability to utilize nitrogen due to the absence of the gene *crnA* or producing the metabolites patulin and penicillin based on the molecular level research [14]. Considering the diversity of active metabolites, interactions with the host, potential biomedical effects of *P. variabile*, and limited whole genome resources, we sequenced and assembled the first chromosome-level genome of *P. variabile* HXQ-H-1 in this study. The comprehensive, continuous genome could provide a new insight to the significant effects on the survival, growth, and fitness of mangroves and the biosynthesis mechanism of *P. variabile*.

## 2. Materials and Methods

### 2.1. Sampling and Isolation

The fungal strain HXQ-H-1 was isolated from the mangrove rhizosphere soil collected on the coast of Fujian Province (Fuzhou, China) and cultivated on modified martin solid plates with hygromycin 48 h, till forming the moderate clones. It was identified as *P. variabile* based on sequencing of the ITS region (GenBank no. KT429657) with 100% similarity to *P. variabile*. The strain was deposited at the Key Laboratory of Marine Drugs, the Ministry of Education of China, School of Medicine and Pharmacy, Ocean University of China, Qingdao, China.

### 2.2. DNA Extraction and WGS Library Construction

The fungi DNA was extracted using CTAB method and sheared into fragments between 100 and 500 bp in size by Covaris E220 ultrasonicator (Covaris, Brighton, UK). High-quality DNA were selected using AMPure XP beads (Agencourt, Beverly, MA, USA). After repairing using T4 DNA polymerase (Enzymatics, Beverly, MA, USA), the selected fragments were ligated at both ends to T-tailed adapters and amplified using KAPA HiFi HotStart ReadyMix (Kapa Biosystems, Wilmington, NC, USA). Then amplification products were subjected to a single-strand circularization process using T4 DNA Ligase (Enzymatics) and generated a single-stranded circular DNA library.

### 2.3. High-Through Chromosome Conformation Capture (Hi-C) Library Construction

Fresh fungi were cut into 10 mm fragments and resuspend in cold 1× phosphate buffered saline (PBS). Then 37% formaldehyde were added to fix the crosslinked to obtain 1% final concentration, and quenched the reaction by adding 2.5 M glycine solution to a final concentration of 0.2 M. The fungi were ground to fine powder in liquid nitrogen. Cross-linked nuclei isolated after centrifugation. Then DNA was digested by the restriction enzyme (Mbo I) (NEB, Ipswich, MA, USA), followed by repairing the 5′ overhang using a biotinylated residue, and blunt-end fragments were ligated in situ using T4 DNA ligase. DNA was reverse-crosslinked and purified using a CTAB method. Finally, Hi-C library was created by shearing of DNA and capturing the biotin-containing fragments with streptavidin-coated beads using Dynabeads MyOne Streptavidin T1 (Invitrogen, Waltham, MA, USA). After DNA fragments end repairing, adenylation, and adaptor ligation, the fragments were amplified by PCR (8 cycles). A single-strand circularization process was carried out as WGS library described.

### 2.4. Genome Sequencing and Assembly

The WGS and Hi-C library were loaded and sequenced on the BGISEQ-500 platform [15]. Raw data are available in the CNGB Nucleotide Sequence Archive under the accession number CNP0000487. (CNSA: https://db.cngb.org/cnsa). The raw reads with high ratio of N (ambiguous base) and low-quality base were filtered out using SOAPnuke (v1.5.2) [16] with parameters “-l 20 -q 0.2 -n 0.05 -Q 2 -c 0 -d”. Then the WGS clean data was assembled using SOAPdenovo (v2.0.4) [17] with parameters “-K 77 -d 1 -R”. Subsequently, the Hi-C reads were mapped to the contigs of draft assembly using HiC-Pro (v2.8.0_devel) [18] with default parameters and proper mapped reads were extracted as the valid Hi-C reads. Then, the draft genome was anchored into chromosomes using 3D-DNA pipeline (v170123) [19] with parameters “-m haploid -s 4” according the interactive positions of Hi-C valid reads. BUSCO (v3.0.1) [20] was used to assess the confidence of the assembly with pezizomycotina_odb9. Simultaneously, we aligned the assembly sequences to the genome of *T. islandicus* WF-38-12 using lastz (v1.02.00) [21] with parameters “T = 2 C = 2 H = 2000 Y = 3400 L = 6000 K = 2200” and visualized using Circos (v0.69) [22].

### 2.5. Identification of Repetitive Elements and Non-Coding RNA Genes

Repetitive sequences were identified using multiple tools. TEs were identified by aligning against the Repbase [23] database using RepeatMasker (v4.0.5) [24] with parameters “-nolow -no_is -norna -engine wublast” and RepeatProteinMasker (v4.0.5) with parameters “-noLowSimple -pvalue 0.0001” at DNA and protein levels respectively. Meanwhile, de novo repeat library was detected using RepeatModeler (v1.0.8) and LTR-FINDER (v1.0.6) [25] with default parameters. Based on the de novo identified repeats, repeat elements were classified using RepeatMasker (v4.0.5) [24] with the same parameters. Furthermore, the tandem repeats were identified using Tandem Repeat Finder (v4.07) [26] with parameters “-Match 2 -Mismatch 7 -Delta 7 -PM 80 -PI 10 -Minscore 50 -MaxPeriod 2000”.

For non-coding RNA (ncRNA), the tRNA genes were predicted using tRNAscan-SE (v1.3.1) [27] with default parameters. The rRNA fragments were identified using RNAmmer (v1.2). The snRNA and miRNA genes were predicted using CMsearch (v1.1.1) [28] with default parameters after aligning against the Rfam database [29] with blast (v2.2.30).

### 2.6. Gene Prediction and Function Annotation

For homolog gene prediction, 4 species protein sequences including *P. Expansum* (GCF_000769745.1), *P. rubens* (GCF_000226395.1), *T. marneffei* (GCF_000001985.1) and *T. stipitatus* (GCF_000003125.1) were download from NCBI for homology-based prediction. These protein sequences were aligned to *P. variabile* assembly sequences using BLAT (v36) [30] and for each protein 5 best matches were retained. Then, gene models were predicted using GeneWise [31] according the aligned loci with default parameters. For de novo prediction, Augustus (v2.4.1) [32] (v3.1) and Genemark (v4) [33] were used with parameters “--ES –fungus”. Finally, all the predicted gene models were integrated using GLEAN [34] with parameters “--minlen 150 --minintron 11 --maxintron 10000” to obtain a consensus gene set. The gene set was assessed using BUSCO (v3.0.1) [20] with pezizomycotina_odb9 in protein mode.

The predicted genes were aligned to the KEGG [35], Swissprot [36], TrEMBL [37], COG [38], CAZy [39], NR and NT [40] database using Blast (v2.2.26) [41] with *E*-value < 1 × 10^−5^. Meanwhile, gene motifs and domains were identified against multiple protein databases including Pfam [42], SMART [43], PANTHER [44], PRINTS [45], ProSite [46] and ProDom [47] using InerProScan (5.16–55.0) [48], and Gene Ontology (GO) [49] terms were obtained based on the InterPro entries. The *P. variabile* assembly was uploaded to antiSMASH (v5.0) [50] website to identify the secondary metabolite gene cluster.

### 2.7. Core Genes, Homolog Genes, SNPs and Phylogeny Reconstruction

Gene sets of 33 *Penicillium* genomes published in NCBI, together with our strain were clustered using Cd-hit (v4.6.6) [51] with parameters “-c 0.5 -n 3 -p 1 -g 1 -d 0 -s 0.7 -aL 0.7 -aS 0.7” to generate a pan gene set, and the clusters present in all strains were extracted as a core gene set. To identify the homolog genes, all protein sequences were aligned to each other using blast+ (v2.2.31) and eliminated the redundancy by solar (v0.9.6). Clustering of the gene family was carried out by the alignment results with hcluster_sg (v0.5.1). Multiple sequences alignment of core genes and clustered gene families was performed by MUSCLE (v3.8.31) [52]. The SNPs among the 34 genomes were detected. The 33 downloaded genomes were aligned to the repeat marked reference genome HXQ-H-1 respectively using MUMmer (v3.23). The SNPs from each sample were aligned follow by reference locations. Phylogenetic trees were established by treebest (v1.9.2) with phyml [53] method and the number of bootstraps was 1000.

## 3. Results and Discussion

### 3.1. Genome Assembly

We obtained 20.24 Gb NGS clean data after quality control (Appendix A). The genome size of *P. variabile* was estimated to be 35.56 Mb according to 17 bp k-mer frequency distribution (Appendix A). The draft genome was assembled with total length of 34.06 Mb, contig N50 of 94.65 kb (Table 1), closing to the estimated size. Hi-C sequencing produced about 112.76 Gb high quality data (Appendix A). We used 1.70 Gb valid data to anchor the draft assembly into seven chromosomes. The final chromosome-level assembly consists of 348 scaffolds with total length of 33.32 Mb, scaffold N50 of 5.23 Mb (Table 1). The longest seven chromosome-level scaffolds comprised 95.91% of the final assembly (Figure 1a,b).

To assess the quality of assembly, firstly WGS clean data were aligned back to the final assembly and a total of 97.93% reads were mapped. Based on the GC-depth distribution (Figure 1c), the GC content was concentrated in 40–55%, and the depth distribution was concentrated in 120–160, without GC or depth bias. Secondly, 3090 (97.9%) single-copy genes were found completely (Figure 1d) according to Benchmarking Universal Single-Copy Orthologs (BUSCO) assessment. The above analysis results indicating the reliability of our genome assembly and can support the downstream analysis.

Here, we assembled the high-quality genome of *P. variabile* into chromosomes using WGS sequences combined with Hi-C sequencing data. Though Hi-C technology has been frequently used in the research of various animal and plant genomes, and assisted assembly to obtain chromosome-level genomes and high-resolution chromatin three-dimensional structural information, it had not been reported in fungal genomes [54]. We constructed the fungal Hi-C library using the same experimental method as terrestrial plant Hi-C libraries [55]. It suggested that the Hi-C sequence method has a great boost to the auxiliary assembly of the fungi genome. With the decline in the cost of next generation sequencing, this strategy can be applied to other fungal species to improve the assembly qualities.

### 3.2. Genome Annotation

We combined the methods of de novo and Repbase to identify repetitive sequences in HXQ-H-1 genome. About 0.9819% (~327,160 bp) and 1.1729% (~390,774 bp) of the genome sequences were detected as repetitive sequences using de novo method and database respectively. In total, the repetitive sequences account for 1.88% (~627.53 kb), including 89.60 kb of DNA repeat elements (DNA), 52.67 kb of long interspersed nuclear elements (LINE), 2.88 kb of short interspersed nuclear elements (SINE), 93.08 kb of long terminal repeat elements (LTR) and 172.04 kb of tandem repeats (TRF) (Appendix A). 

Furthermore, we predicted 10,622 protein-coding genes in all, and there were 9927 *T. islandicus* WF-38-12 and 9000 *P. digitatum* [13,14]. For these 10,622 protein-coding genes, the average length of genes, coding sequences, exons, and introns were 1994.87 bp, 1568.29 bp, 473.94 bp, and 184.74 bp, respectively (Table 2). The gene set was assessed using BUSCO [20] with pezizomycotina_odb9 database, and 2788 (88.3%) out of 3156 single-copy orthologs can be found completely, indicating the high quality of gene set (Figure 1d). 

Noncoding RNAs (ncRNAs) have complex regulatory functions at the transcriptional, translational and epigenetic level in fungi [56]. We predicted a total of 324 ncRNAs in HXQ-H-1, including 92 tRNAs, 3 rRNAs, 63 sRNAs, 50 snRNAs and 115 miRNAs (Appendix A). Compared to the related species *T. islandicus*, the number of tRNAs was smaller (107 in WF-38-12) [13].

### 3.3. Functional Categorization

Of the 10,622 predicted genes, a total of 10,586 genes (99.66%) were functionally annotated by 9 databases (Figure 2a). In KEGG annotation result, 8201 genes (77.21%) were annotated and classified into 130 pathways belonging to 24 Level2 (Figure 2b). For COG functional annotation, 5574 genes (52.48%) were annotated to the 24 COG categories (Appendix A). The most abundant cluster was related to the metabolism, including “amino acid transport and metabolism” and “carbohydrate transport and metabolism”. Further, 5533 genes were classified according to the GO database, in which 8359, 8025 and 5679 genes were annotated to “cellular component”, “biological process” and “molecular function”, respectively. The “catalytic activity” was the largest annotated GO term (Appendix A).

Furthermore, there were 654 (6.16%) genes were annotated by CAZymes. Among these genes, 243 genes encoding glycoside hydrolases (GHs) accounted for the largest proportion (2.29%). In addition, 134 genes were belonged to glycosyltransferases (GTs) (1.26%) (Figure 2c). Compared with the other 33 *Penicillium* genomes (Appendix A), we found that HXQ-H-1 had the highest gene number in AA4, CBM35, CBM56, GH54, GH76, GT21, and PL7. Notably, CBM56 was annotated uniquely in HXQ-H-1 (Appendix A), which may provide hints to understanding the energy flow and manglicolous fungus’s lifestyle.

The secondary metabolism is an important characteristic of fungi and display a dichotomy within biological activities, some benefit the society by developing into medicines and agrochemicals, and some are pathogenic determinants of humans, animals and plants [57,58]. HXQ-H-1 secondary metabolites extractions such as nafuredin B showed cytotoxicity against a panel of human cancer cell lines, as well as varitatin A showed cytotoxicity against HCT-116 cell line (human colon cancer) and also inhibited the effects of protein tyrosine kinases [6,7]. To observe secondary metabolites biosynthetic gene clusters in HXQ-H-1, we further annotated the genome with antiSMASH pipeline [50]. A total of 77 gene clusters were detected (Appendix A), including 29 polyketide synthase (PKS), 23 non-ribosomal peptide synthetase (NRPS), 10 mixed (NRPS-PKS, PKS-NRPS-NRPS like-terpene, T1PKS-NRPS like, T1PKS-terpene), eight terpene (Terpene), three indole cluster (indole), one β-lactone containing protease inhibitor (β-lactone), one Fungal RiPP with POP or UstH peptidase types and a modification (fungal-RiPP), one siderophore (siderophore), and one biosynthesis clusters (other). These results show more secondary metabolic clusters than those in *T.islandicus* [13] and the genes indole, betalactone, fungal-RiPP, and siderophore are absent in *T. islandicus*. It has been reported that most drug or drug-like metabolites are PKS, NRPS and terpene synthase and these enzymes have been extensively annotated in HXQ-H-1, reflecting that HXQ-H-1 has the large capacity of biotechnological potential as novel drugs [58].

Due to the special ecologic niche of mangroves, the ability of secondary metabolites of microorganisms is given to conquer harsh habitats. Most biosynthetic gene clusters of secondary metabolites are silent under standard laboratory culture conditions, new methods are needed to activate gene expression for metabolism exploration [59]. Our genomic analysis results here can provide guidance for gene regulation and facilitate the further studies of secondary metabolites in *P. variabile*.

### 3.4. Genome Comparison of P. variabile and T.islandicus

The genome collinearity between *P. variabile* HXQ-H-1 and *T. islandicus* WF-38-12 was highly consistent with 61.75% colinear ratio in HXQ-H-1. Our 7 chromosomes (scaffold 1–7) were primary matched to 13 chromosomes (CVMT01000001.1- CVMT01000013.1) of strain WF-38-12 (Figure 3). In detail, scaffold 1–6 corresponded to multiple scaffolds (scaffold1 vs. T01000003.1, T01000005.1, scaffold2 vs. T01000008.1, T01000001.1, scaffold3 vs. T01000013.1, T01000006.1, T01000004.1, T01000002.1, scaffold4 vs. T01000002.1, T01000012.1, scaffold5 vs. T01000002.1, T01000007.1, T01000011.1, scaffols6 vs. T01000010.1, T01000009.1). Scaffold 7 involved part of T01000001.1 (scaffold7 vs. T01000001.1). The GC content of HXQ-H-1 (47.59%) is similar to WF-38-12 (46.25%), and gene density are distributed without obvious bias along chromosomes (Figure 3). Through the COG annotation heatmap, we find that the percentage of HXQ-H-1 genes annotated by COG is relatively higher than WF-38-12 (Figure 3). Interestingly, there is a small difference in the total gene number (10,622 vs. 9927), whereas there is a relatively large difference in gene number with COG functional annotation (5574 vs. 5012). The main different functional categories were “signal transduction mechanisms”, “coenzyme transport and metabolism”, “energy production and conversion”, “lipid transport and metabolism”, and “secondary metabolites biosynthesis, transport, and catabolism”.

### 3.5. Phylogenetic Analysis

To determine the HXQ-H-1 relationship among members of *Penicilliun*, we next reconstructed the species phylogenetic tree based on the whole genome of *P. variabile* HXQ-H-1 and other 33 *Penicillium* species (Appendix A). We reconstructed trees using three kind of datasets using their core genes, homolog genes and SNPs, respectively. We obtained 582 core genes, 82 homolog genes and 3597 SNPs. All three phylogenetic trees are well-supported with boostrap values more than 95% (Figure 4). They are generally congruent and could be obviously divided into three clades. Previous studies reported the *Penicillium* was polyphyletic and clade2 were *Talaromyces* species described as a sexual state of *Penicillium* [60,61]. In our results, the *P. variabile* HXQ-H-1 is mostly close to the *T. islandicus* WF-38-12, consistent with previous phylogenetic tree construction using ribosomal DNA [62,63]. There are 48.18% (5118) genes in strain *P. variabile* HXQ-H-1 are orthologous with *T. islandicus* WF-38-12. It’s the first time *Penicillium* phylogenetic tree reconstructed by genome-wide sequence data. We found that the traditional morphological classification of *Penicillium* species was confusing, leading to polyphyletic classification [60,61]. Biological classification needs to correctly reflect the phylogenetic relationship and evolutionary trends. With the development of molecular phylogenetics, the fungal classification has undergone tremendous transform. The reconstructed phylogeny three here may provide a reference for phylogenetic assessment or population genetics research.

## 4. Conclusions

Mangrove is a special tropical coastal vegetation between the ocean and interior, an area home to a large number of marine fungi and terrestrial fungi. Manglicolous fungi plays an important role in mangrove ecosystems as a decomposer, and its metabolites not only help mangroves fight disease, but also help mankind to exploit new drugs. However, the research on mangrove fungi is not extensive. According a white paper, only 339 manglicolous fungal species have been identified and 46 marine (contain mangrove) fungi genome assembled [64]. As far as we know, this is the first *Penicillium* genome in mangrove ecosystems. Based on this assembly, we identified abundant repetitive sequences, protein-coding genes, and ncRNAs. According to different evaluation results, our high-quality genome assembly and accurate protein-coding genes prediction were also confirmed. We performed gene functional annotation including secondary metabolism, which reveals the large capacity, biotechnological potential of *P. variabile* HXQ-H-1 based on molecular biology. We created out genome-wide species phylogeny trees of *Penicillium* for the first time. All the resources provided here are convenient and helpful for comparative genomic analysis, phylogenetic analysis, and metabolic regulation analysis in the future.

## Figures and Tables

**Figure 1 jof-06-00007-f001:**
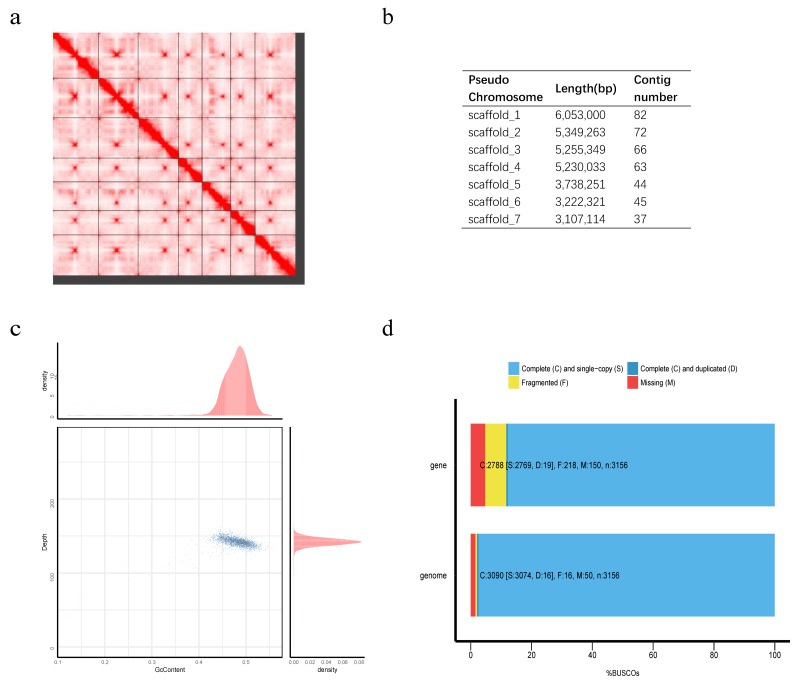
Summary of genome assembly and assessment. (**a**) The seven longest scaffolds interactive heatmap of *Penicillium variabile* assembly. The horizontal and vertical lines divide the genome into seven major pseudochromosomes. Gradient red indicates the intensity of contact between sequences. The intra-contact of sequences in pseudochromosome is stronger than the inter-contact. (**b**) The length of each chromosome and involved contig number. (**c**) GC-depth distribution of final assembly. The horizontal coordinate represents the average GC content per 10 kb windows, and the vertical coordinate represents the corresponding coverage depth. The bar charts on the top and right represent the distribution of GC and coverage depth. (**d**) BUSCO (benchmarking universal single-copy orthologs) assessment of genome and gene set. The bar chart below is a BUSCO assessment of genome, of which 3090 genes (97.91%) are predicted to be complete including 3074 single copies and 16 duplications. The upper bar chart is a BUSCO assessment of gene set, of which 2788 genes (88.34%) are predicted to be complete, including 2769 single copies and 19 duplications.

**Figure 2 jof-06-00007-f002:**
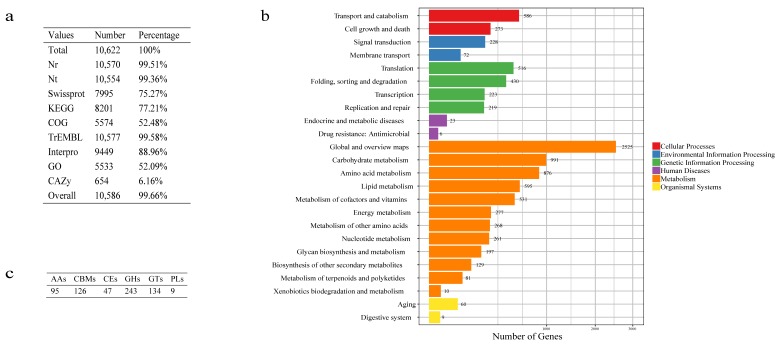
Function annotation of strain *P. variabile* HXQ-H-1 (**a**). Statistics of annotated genes by 9 databases. The third column indicates the ratio of the annotated genes number. (**b**). KEGG pathway classification. Histograms represent the gene number of pathways and grouped into 6 classifications which were tagged by colors. (**c**). Gene numbers of annotated by CAZyme.

**Figure 3 jof-06-00007-f003:**
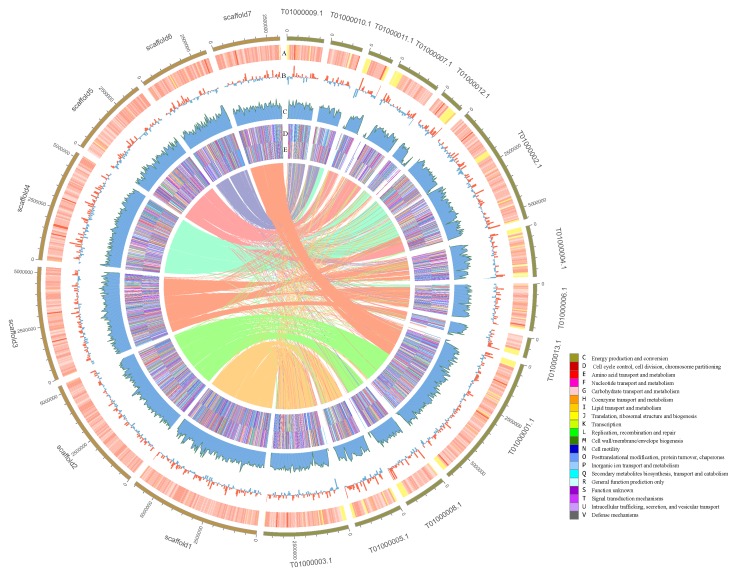
Genome comparison of *Penicillium variabile* HXQ-H-1 and *Talaromyces islandicus* WF-38-12. The lefts are the 7 scaffolds of HXQ-H-1, and the rights are the 13 scaffolds of WF-38-12. The collinear relationship between scaffolds is connected by center thin lines. The scale of each small square of the outermost circle represents 5 kb. Circle A is GC heatmap. Circle B is the histogram of GC (red: G > C; blue: G < C). Circle C is gene density in chromosomes. Circles D and E are the COG positive/negative annotation heatmaps, and the legend is shown in the bottom right.

**Figure 4 jof-06-00007-f004:**
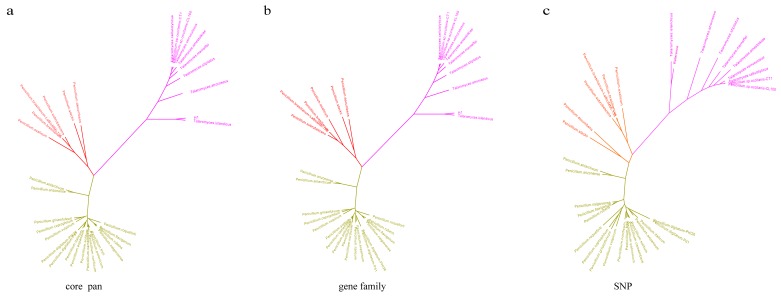
The species phylogenetic trees of *Penicillium variabile* HXQ-H-1 and other 33 *Penicillium* genomes. Trees were reconstructed based on core genes (**a**), homolog genes (**b**) and SNPs (**c**) respectively. The three clades are marked with different colors.

**Table 1 jof-06-00007-t001:** Statistic of genome assembly.

Values	Draft Genome		Chromosome Genome	
	Scaffold	Contig	Scaffold	Contig
Num	3860	3860	348	757
Length (bp)	34,059,084	34,059,084	33,318,899	33,114,399
N50 (bp)	93,482	93,482	5,232,202	96,738
N90 (bp)	24,393	24,393	2,951,000	30,747
GC%	46.58	47	48	48

**Table 2 jof-06-00007-t002:** Gene prediction results using homologous and de novo.

Values	Homolog				*denovo*	Glean
	*P. expansum*	*P. rubens*	*T. marneffei*	*T. stipitatus*	Augustus	
Gene_number	8947	9214	9510	9360	12,032	10,622
Average_gene_len (bp)	1456.62	1394.67	1459.34	1478.38	1773.88	1994.87
Average_cds_len (bp)	1317.95	1294.49	1349.26	1363.3	1581.44	1568.29
Average_exon_number	2.34	2.31	2.43	2.47	3.38	3.31
Average_exon_len (bp)	562.45	560.76	554.42	552.59	468.3	473.94
Average_intron_len (bp)	103.24	76.57	76.78	78.44	80.96	184.74

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
