# Peer review of "Chromosome-Level Comprehensive Genome of Mangrove Sediment-Derived Fungus Penicillium variabile HXQ-H-1"

_jof, 2019, doi:10.3390/jof6010007_

Round 1

Reviewer 1 Report

The topic covered in the paper is certainly interesting and worthy of being published.

However, I have a doubt concerning the choice of ITS as molecular marker for the identification of P. variable as the preferred marker for the identification of Penicillium species is Beta tubulin.

I also wonder if the studied organism is kept in a public access and recognized collection of microorganisms. If yes, please indicate which one.

I also point out some form errors that should be fixed before publication.

- The name of the organisms must be written in full the first time it is mentioned in the paper, from the second time onwards, for example P. variable. Firstly, my doubts concern the choice of ITS as a molecular marker for the identification of P. variable as the preferred marker for the identification of Penicillium is Beta tubulin. I also wonder if the studied lineage is kept in a public access and recognized collection.

I also point out some form errors that should be solved before publication.

The name of the oragnisms must be written in full the first time it is mentioned in the paper and in the abbreviated form (e.g. P. variabile) from the second time onwards. The current name of Talaromyces marneffei is Penicllium marneffei.

Author Response

We want to thank Reviewer #1 for careful review and insightful criticism and advice. Below is our point-by-point response to the reviewer’s comments:

Point 1: However, I have a doubt concerning the choice of ITS as molecular marker for the identification of P. variable as the preferred marker for the identification of Penicillium species is Beta tubulin.

Response 1: The ITS conserved sequence and Beta tubulin are the molecular characteristic sequences commonly used in fungal identification. Nuclear ribosomal internal transcribed spacer (ITS) region is a universal DNA barcode marker for Fungi (doi.org/10.1073/pnas.1117018109), and we found ITS has also broadly used for the identification of other Penicillium species. We do not test the tubulin again due to that the strain is a common fungus and the similarity of ITS sequence is high enough.

Point 2: I also wonder if the studied organism is kept in a public access and recognized collection of microorganisms. If yes, please indicate which one.

Response 2: The strain was deposited at the Key Laboratory of Marine Drugs, the Ministry of Education of China, Ocean University of China, Qingdao, People’s Republic of China, which is one of the Ministry of Education Key Laboratories of China. The strain HXQ-H-1 is access, any researcher can email to the corresponding author to request for the strain.

Point 3: I also point out some form errors that should be fixed before publication.

- The name of the organisms must be written in full the first time it is mentioned in the paper, from the second time onwards, for example P. variable. Firstly, my doubts concern the choice of ITS as a molecular marker for the identification of P. variable as the preferred marker for the identification of Penicillium is Beta tubulin. I also wonder if the studied lineage is kept in a public access and recognized collection.

Response 3: Thank you for pointing the name of the organisms out, we have corrected it in the paper. For the ITS marker and access issue, please see the responses of the questions above.

Point 4: I also point out some form errors that should be solved before publication.

The name of the oragnisms must be written in full the first time it is mentioned in the paper and in the abbreviated form (e.g. P. variabile) from the second time onwards. The current name of Talaromyces marneffei is Penicllium marneffei.

Response 4: For the name of Talaromyces marneffei, we used the species name according to the information of corresponding NCBI accession number. (https://www.ncbi.nlm.nih.gov/assembly/GCF_000001985.1/). We also find a literature supporting Talaromyces marneffei (doi: 10.3114/sim.2011.70.04).

Reviewer 2 Report

L. Peng et al. present an interesting report on the effort to sequence and annotate the complete genome of Penicillium variabile and its comparison with other available genomes on the genus. Making available the results of this study will help the scientific community in the characterization of fungi molecular processes, their ecological role and adaptations to different ecosystems. However, much needs to be improved to maximize the potential impact of this study.

English proof-reading of the manuscript is needed to deliver the intended message. Some examples of confusing phrases through the document:

-line 42: "consist" should be changed to "represent"

-line 194: "litter higher", maybe it was intended to say "little higher", which should be also avoided in favor of "higher".

-line 255: "we can find that HXQ-H-1 has higher abundant", it is not understandable.

On the other hand, some major points to correct through the manuscript:

-How was T. islandicus identified as the closest species to P. variabile? As the last section presents the phylogenetic analysis, the conclusion of T. islandicus as closest species seems out of order because its genome was already used to assemble the genome of P. variabile. In other words, at least the ribosomal DNA should be mention earlier (in the Introduction) as a reason to choose using T. islandicus as the base to assembly the P. variabile. Also, the T. islandicus is a draft genome, therefore the interpretation of the number of genes and presence/absence of genes should be considered with some caution.

-In multiple occasions through the text there are mentions of bioactive antibiotics, drug candidates and biotechnological applications; however, little to none examples are provided. There is a mention that "The versatile secondary metabolites of P.variabile HXQ-H-1 reflect large capacity of biotechnological potential as novel drugs according to the literature."; however, no paper or book on that literature is mentioned. Also, given the importance of such a claim, examples of genes identified in such pathways could be useful. Or at least some speculation on the role that they perform in the ecological niche.

Round 2

Reviewer 2 Report

The authors modified the text satisfactorily and the contents have improved significantly.